# A PCB Alignment System Using RST Template Matching with CUDA on Embedded GPU Board [note 1]

**DOI:** 10.3390/s20092736

**Published:** 2020-05-11

**Authors:** Minh-Tri Le, Ching-Ting Tu, Shu-Mei Guo, Jenn-Jier James Lien

**Affiliations:** 1Department of Computer Science and Information Engineering, National Cheng Kung University, No. 1 University Road, Tainan City 701, Taiwan; n28057023@mail.ncku.edu.tw (M.-T.L.); guosm@mail.ncku.edu.tw (S.-M.G.); 2Department of Applied Mathematics, National Chung Hsing University, No. 145, Xingda Road, Taichung City 402, Taiwan; cttu@dragon.nchu.edu.tw

**Keywords:** alignment system, PCB manufacturing, template matching, embedded system, GPU, parallel programming

## Abstract

The fiducial-marks-based alignment process is one of the most critical steps in printed circuit board (PCB) manufacturing. In the alignment process, a machine vision technique is used to detect the fiducial marks and then adjust the position of the vision system in such a way that it is aligned with the PCB. The present study proposed an embedded PCB alignment system, in which a rotation, scale and translation (RST) template-matching algorithm was employed to locate the marks on the PCB surface. The coordinates and angles of the detected marks were then compared with the reference values which were set by users, and the difference between them was used to adjust the position of the vision system accordingly. To improve the positioning accuracy, the angle and location matching process was performed in refinement processes. To overcome the matching time, in the present study we accelerated the rotation matching by eliminating the weak features in the scanning process and converting the normalized cross correlation (NCC) formula to a sum of products. Moreover, the scanning time was reduced by implementing the entire RST process in parallel on threads of a graphics processing unit (GPU) by applying hash functions to find refined positions in the refinement matching process. The experimental results showed that the resulting matching time was around 32× faster than that achieved on a conventional central processing unit (CPU) for a test image size of 1280 × 960 pixels. Furthermore, the precision of the alignment process achieved a considerable result with a tolerance of 36.4 μm.

## 1. Introduction

In the assembly line manufacturing of electronic devices, the PCB alignment process plays a critical role in pinpointing the positions of the components, checking for missing integrated circuits ICs or devices, carrying out defect inspections and performing soldering. The alignment procedure demands efficient and high-accuracy processing and is most commonly performed by using some form of machine vision system to detect and locate the positions of the fiducial marks and components on the printed circuit board (PCB) [1,2,3,4]. Ideally, the PCB alignment system should have both a small size and a low cost in order to enhance the flexibility and reduce the total manufacturing cost, respectively.

On the other hand, embedded vision systems, consisting of a camera directly integrated with a processing board, have a small size, good portability and a low power consumption. As a result, they are nowadays widely applied in many practical applications, including object recognition, tracking, medical image processing, automatic car driving and defect inspection systems [5,6,7,8,9,10,11]. However, they suffer the disadvantages of a long processing time due to their limited central processing unit (CPU) capability and a relatively small memory space. Consequently, the problem of accelerating the processing time of embedded vision systems has attracted significant attention in recent years. The authors in [12,13,14,15,16] attempted to improve the performance of the embedded systems through the use of enhanced algorithms and models. In contrast, applying the development of hardware is also taken into consideration. The studies in [17,18,19,20,21,22] used hardware-based techniques to improve the performance of the embedded systems by running the image-processing algorithms in parallel on graphics processing units (GPUs) or field-programmable gate arrays (FPGAs). The present study constructed a PCB alignment algorithm based on a rotation, scale and translation (RST) template matching technique. The algorithm first detected the fiducial marks on the PCB and then compared the detected values of the mark coordinates and angles with the corresponding reference values. The difference between the two sets of results was then used to align the vision system with the PCB. To satisfy the requirements for a small size, low-cost and good portability, the entire RST algorithm was implemented on a GPU-embedded system (NVIDIA Jetson Tx2 development kit, NVIDIA Corporation, Santa Clara, CA, United States).

Template matching is one of the most commonly used machine vision techniques. It uses a template to scan across an image and uses a similarity measurement method to detect the required targets. The template matching has been widely applied in many research areas, including object detection, tracking and pattern recognition. The main advantage of template matching is to enable the outputs of the matching process (i.e., the coordinates, angle or scale of the detection target) to be obtained with an extremely high degree of precision. Moreover, the template matching process requires only a small number of data samples for training purposes and can be performed using only a single template. Nevertheless, the template matching is an expensive technique. The first reason for that is the similarity measurement methods. Normalized cross correlation (NCC) is one of the most commonly used methods. In comparison to other similarity formulae such as the sum of square differences (SSD) or the sum of absolute differences (SAD), the NCC formula is more robust toward variations in the brightness and contrast conditions. However, the NCC formula incurs a high computational cost since, to achieve high-precision scale and rotation matching, it is necessary to use more NCC formulae for comparing. As a result, the matching time is inevitably increased. Secondly, in performing the matching process, scanning time is another considerable problem. The larger the image is, the slower the scanning time is. Finally, the computation time of the matching algorithm is rather long due to the limited processing capability of the embedded CPU on which it generally runs.

To address these limitations, in this article we proposed a PCB alignment system in which the RST template matching process was performed with compute unified device architecture (CUDA) on an embedded GPU board. There were three major contributions as follows: (1) the number of similarity measurements was reduced by quickly rejecting weak features in the scanning process. From that, we can cut down the matching time; (2) moreover, in this step, the NCC formula was replaced by a sum of products to further reduce the number of operations in the rotation matching process; (3) furthermore, we applied hash functions to find refined positions in the refinement matching process when reducing the scanning time by applying the implement of matching processes on the parallel threads of an embedded GPU.

The remainder of this paper is organized as follows. In Section 2, we present researches which were related to our work. The framework of the proposed PCB alignment system is presented in Section 3. Section 4 describes the proposed RST template matching. The next Section is used for the description of how we accelerated the RST algorithm. Then, Section 6 presents and discusses the experimental results. Finally, we provide some brief concluding remarks and indicates the intended direction of future research in Section 7.

## 2. Related Work

Reducing the matching time in template matching is one of the key considerations in machine vision. Kim et al. [23,24] performed RST template matching using circular and radial features to deal with the scale and angle of the targets. In addition, to improve the matching time, fast Fourier transformation was applied to reduce the computational cost of the NCC formula. Hsu and Shen [25] presented a method for checking the integrated circuit (IC) marking on an embedded platform automatically by first detecting the angle of the chip and then using the SAD similarity measure to match the template image and target image. To reduce the scanning time in the inspection process, the detection algorithm was implemented on a multi-core embedded processor. Moreover, the operations of the algorithm were performed in parallel using Single Instruction Multiple Data (SIMD) instructions for Arm processors known as NEON. Annaby et al. [26] proposed an improved template matching technique for detecting missing components in PCB manufacturing. In the proposed method, the computation time of the NCC matching process was reduced by converting 2D blocks of images into 1D blocks and then applying discrete cosine transformation (DCT) to the NCC formula. Moreover, to deal with the matching time of template matching, Shih et al. [27] developed a robust and rapid template matching method designated as Spiral Aggregation Map (SPLAM) in which scale-insensitive spirals were used to extract the features required to perform rotation matching. The authors dealt with matching time by applying the coarse-to-fine approach into position and orientation matching. Firstly, a coarse matching process was used for position matching based on a single angle. Then, from the results, they carried out the angle matching with more angles. Lai et al. [28] presented a method for improving the acceleration and accuracy of the template matching process by using annulus projection transformation (APT) vectors as image descriptors. The matching process was accelerated by compressing a 2D image into a 1D vector. Chen et al. [29] proposed a pyramid hierarchical strategy to search the matching position. The authors used the Hough transform to find the initial angle. After that, refinement matching was used to improve the accuracy of the position and the orientation matching process. Wu and Toet [30] converted blocks within the template and target images into weak binary features. To speed up the matching time, the authors used integral images to calculate weak binary blocks. In addition, they fast rejected poor matching patterns by using cascaded computation. Cai et al. [31] presented an improved template matching approach for real-time visual servoing systems, in which a normalized SAD method was used to detect the target by subtracting the gray values of the template image from those of the reference image and then dividing the result by the maximum gray value to obtain a normalized score, where a maximum score indicated an improved matching outcome. To improve the scanning time, firstly, a coarse matching process was scanned with a large step to obtain the initial matching positions. Then, the matching process was carried out in square regions which took the initial matching position as the center and the large step of the coarse matching process as the side length. Liu et al. [32] presented a method for reducing the time of rotation- and scale-invariant template matching by rapidly ruling out the regions of no possible matches. The computational cost was further reduced by applying the summed area table and it was also computed on a parallel computing architecture. To deal with the rotation and scale invariant matching, the authors used octagonal-star-shaped templates and scaled them with different sizes. Then, those scaled templates were compared with reference image to find the matching targets.

The use of GPUs to accelerate the template matching process has attracted significant attention in the recent literature. Rakvic et al. [33] presented a template matching-based iris recognition algorithm, in which segmented iris images were converted into binary images and a matching process was then performed by comparing the binary images with template images stored in a database using the Hamming distance measure. To reduce the matching time, the segmentation process and matching process were both embedded on a GPU in order to parallelize their computation. The authors in [34,35] proposed template matching processes based on sliding windows for performing rapid earthquake detection on a GPU. Yan et al. [36] presented an effective parallel strategy to speed up the matching process on the thread blocks of a GPU which runs on a NVIDIA Jetson TX2. It was shown that the resulting matching time was almost six times faster than the selected comparison methods. Li and Pan [37] accelerated the speed of the image matching process in binocular visual registration processes using CUDA code and a scale-invariant feature transform (SIFT) algorithm. As a result, the matching time on the GPU was more than 7.2 times faster than on CPU.

## 3. Proposed Embedded PCB Alignment System

The proposed embedded PCB alignment system was a combination of two components: hardware and software. The former consisted of a PCB alignment platform and a NVIDIA Jetson Tx2 development kit, while the latter included a proposed RST template matching algorithm and a GPU parallel computation with C/C++ using CUDA. 

### 3.1. System Hardware Consists of Two Subsystems

In general, system hardware is composed of two subsystems: namely an embedded-based controller and a vision alignment platform. The former was built on a NVIDIA Jetson TX2 development kit (see Figure 1e) incorporating a quad-core 2.0 Ghz 64-bit ARMv8 A57, a dual-core 2.0 Ghz ARMv8 Denver, a 256 CUDA core (1.3 Mhz NVIDIA Pascal) and 8 Gb memory. As shown in Figure 1a–c, the vision alignment platform mainly consisted of three two-phase stepper motors (M1, M2 and M3) and two Basler acA3800-10gm area scan cameras (C1, C2) mounted on a movable platform (PL). The cameras were industrial monochrome cameras with a frame rate of 10 frames per second (fps) at 10 MP resolution and are nowadays widely used in many machine vision, medical, microscopy and factory automation systems.

During the alignment process, the three stepper motors were used to control the motion of the platform with the mounted cameras. In particular, M3 moved the platform along the *y* axis direction, while M1 and M2 adjusted its rotational position. The movement of the platform in the *x* axis direction was achieved by assigning the same motion values to motors *M1* and *M2*, respectively. For each motor, the motion distances were computed as follows:dX_1_ = R*cos(θ + θ_x1_) − R*cos(θ_x1_)   (mm),(1)
dX_2_ = −R*cos(θ + θ_x2_) − R*cos(θ_x2_)   (mm),(2)
dY = −R*cos(θ + θ_y_) − R*cos(θ_y_)   (mm),(3)
where *R* is defined and shown in Figure 1d and has a value of *R* = 42.43 mm in the present implementation; *θ_x1_*, *θ_x2_* and *θ_y_* are defined and shown in Figure 1d and have values of *θ_x1_* = 315°, *θ_x2_* = 135° and *θ_y_* = 225° in the present case; *θ* is the desired rotation angle of the platform; and *dX1*, *dX2* and *dY* are the displacement motions of motors *M1, M2* and *M3*, respectively.

### 3.2. The Pixel-to-Metric Units Conversion Based on Four Reference Points of Cross-hair Marks

One of the main challenges in the alignment control problem considered in the present study was that of converting the coordinates of the PCB marks to metric units (millimeters (mm) in the present case). The stepper motors in the hardware system shown in Figure 1 provide a travel distance of 1 mm for every 1600 pulses. In order to determine the number of pixels corresponding to a travel distance of 1 mm, a calibration process was performed in which the platform was moved along the *x* and *y* axis directions, respectively, through a distance of 1600 pulses to four reference points (see Figure 2). The coordinates of the four points were detected using the RST template matching algorithm (we will present that algorithm in Section 4) and the pixel distances between the points were then measured. Finally, the number of pixels per 1 mm distance along the *x* and *y* axis directions (ΔX and ΔY, respectively) were calculated as follows: (4)$$\Delta X=\frac{\sum_{i=1}^{4}\;\Delta x_{i}}{4}\;(\mathrm{pixels}/\mathrm{mm}),$$
(5)$$\Delta Y=\frac{\sum_{i=1}^{4}\;\Delta y_{i}}{4}\;(\mathrm{pixels}/\mathrm{mm}),$$

### 3.3. Procedure of Alignment between Marks on PCB Surface and Set Points in Field of View of Cameras

It should be assumed that we started to do an alignment process with a template matching algorithm which we will present in Section 4. The alignment procedure comprised five main steps, as shown in Figure 3 and described in the following: 

**Step 1. Two input images were captured by the cameras mounted on the movable platform.** Each camera detected one mark on the PCB surface with a resolution of 1280 × 960 pixels (see Figure 3a). Mark 1 was inside a fields of view (FOV) of the left camera (C1) and Mark 2 was inside a FOV of the right camera (C2). Those marks are shown as red cross-hair marks in Figure 3a. The main purpose of the vision system was to align the marks on the PCB with two set points in the fields of view (FOVs) of the two cameras (shown as blue cross-hair marks in Figure 3a). 

**Step 2. The orientation of the PCB marks was detected using a RST template matching.** A RST template matching process was performed to detect the position coordinates and angles of the PCB cross-hair marks (see Figure 3b) For each cross-hair mark, the RST algorithm output the coordinate and orientation angle with a resolution of 0.1°. It should be noted that the high resolution of the angular detection process increased the accuracy of the coordinate matching process and hence improved the overall precision of the alignment system.

Step 3. Orientation synchronization was performed between the set points and the PCB cross-hair marks by adjusting the rotational position of the platform. Based on the angle derived in the previous step, Equations (1)–(3) were used to determine the motions of the three stepper motors required to rotate the platform so that the FOVs of the two cameras were aligned in parallel with the PCB (see Figure 3c).

**Step 4. The coordinates of the cross-hair marks were detected using the RST algorithm.** The distance between the set points and the PCB marks in the *x* and *y* axis directions was determined by comparing the detected coordinates of the marks with the reference coordinates of the camera set points (see Figure 3d).

**Step 5. The marks and set points were aligned by shifting the platform as required.** Using Equations (1)–(3), the displacements required by the three stepper motors to align the marks and set points were converted into an equivalent number of pulses and the motors were then actuated to move the platform in the horizontal and vertical directions as required (see Figure 3e).

## 4. Refinement Algorithm of the Rotation, Scale and Translation (RST) Template Matching

In performing Step 2 (orientation detection) of the alignment process described in Section 3.3, the present study proposed a RST refinement algorithm. The RST refinement algorithm involved three steps, namely one step for training the template and two steps for performing the testing process. In the training process, before being assigned to the RST algorithm, the template and test image were resampled using a pyramid technique to obtain a down-sampled template (*T’*) and down-sampled test image (*I’*), respectively. This step helped reduce the scanning time in the testing process. The levels of the pyramid (*N_P_*) were set as 0, 1 and 2 depending on the size of the template, i.e., ≤ 40 × 40, (40 × 40) ~ (200 × 200), and ≥ (200 × 200) pixels, respectively. In the training process, *Ns* scale templates were produced and the radial features of each template were extracted for rotation-invariant matching. For each scale template, the radial features were obtained by first creating *Nr* radial lines (described as yellow lines on the template image, *I’*, in Figure 4a), where the angle between the adjacent lines was referred to as the angular resolution (α) and has a value of α = 360/*Nr*. The average grayscale pixel value along each radial line was then computed to derive the corresponding radial feature, denoted as *Rq*. When collecting the pixel values for each radial line, the coordinates of the pixels were stored in a look-up table (*LUT_R*) so that when extracting the radial features of the test image, the respective coordinates could be reused in order to locate the pixels more rapidly.

As shown in Figure 4, the procedure of the RST refinement testing first determined the rotation angle of the PCB by rotation matching (Step 1) and then refined the matching results (Step 2). The initial rotation matching process was performed using radial features such as those described in [10,23]. In addition, the accuracy of the matching results was then improved via a further refinement process. This step itself consisted of two smaller steps, namely location refinement and angle refinement.

**Step 1**. **Angle measurement using rotation matching.** In Step 1, the template *T’* was scanned over the test image *I’* from the top-left corner to the bottom-right corner (see Figure 4a). At each position, *Ns* scale search windows were created. For each scale search window, the coordinates stored in *LUT_R* were used to generate *Nr* radial lines, as described above for the training step. The radial features (*Ra*) were calculated by computing the average grayscale pixel value along each radial line. Angle candidates were then determined by rotating the *Nr* elements of *Ra* and for each rotation of the line, measuring the similarity between *Rq* and the rotated *Ra* using the NCC formula shown in Equation (6). For *Nr* radial lines, *Ra* was rotated *Nr* times. The *Ra* having the highest NCC score among all the rotations of *Ra* was then selected. At each template scanning position, *Ns* maximum NCC scores were obtained. The maximum score among all of these scores was chosen and compared with a pre-set radial threshold value (*t*_1_). If the chosen maximum score was greater than this threshold value, the corresponding angle was chosen as an angle candidate and the coordinates corresponding to this candidate were taken forward to the refinement process performed in Step 2 of the testing process.

The normalized cross correlation (NCC) equation has the form:(6)$$\eta_{NCC}=\left|\frac{\sum_{i=1}^{Nr}\left(Rq_{i}-\mu_{Rq}\right)\left(Ra_{i}-\mu_{Ra}\right)}{\sqrt{\sum_{i=1}^{Nr}{\left(Rq_{i}-\mu_{Rq}\right)}^{2}}\sqrt{\sum_{i=1}^{Nr}{\left(Ra_{i}-\mu_{Ra}\right)}^{2}}}\right|\leq 1.0$$
where $\eta_{NCC}$ is NCC score; *Nr* is number of radial lines; $\left(Rq_{i},\;\mu_{Rq}\right)$ and $\left(Ra_{i},\;\mu_{Ra}\right)$ are the intensity average values and means of the radial lines on a template *T’* and a scale search window, respectively. 

**Step 2. Robust accuracy using refinement matching processes.** As described above, Step 1 of the testing process provided a set of candidate angles and corresponding pixel coordinates. In Step 2, the accuracy of the rotation matching process was enhanced by refining both the positions and angles of the selected candidates. As in the previous steps, the refinement process was performed using the down-sampled template *T’* and test image *I’*. In general, down-sampling reduces a high-resolution image to a lower-resolution image and is hence beneficial in reducing the computational cost of the matching process. However, in doing so, it also reduces the attainable precision of the position matching outcome. Moreover, a higher value of the angular resolution α reduced the angle matching time in Step 1, but limited the accuracy of the angle matching result. To address these issues, the refinement step in the test process consisted of two sub-steps, namely a position matching refinement step and an angle matching refinement step, as described in the following. 

**Step 2.1 Robust location measurement using location refinement matching.** After up-sampling each coordinate candidate obtained in Step 1, the $2^{N_{P}}$ neighboring pixels were expanded (as shown in Figure 4c). Every expansion pixel was then taken as the center of a new search window for the location refinement matching step. Note that the search window had the same orientation as that of the corresponding angle candidate in Step 1 and has a size commensurate with the *Ns* scale. The NCC similarity function was then used to measure the correlations between the template (*T*) and the search windows located at each of the expansion pixels. For each expansion position, a total of *Ns* ∗ $2^{N_{P}}$ NCC scores were obtained. The maximum score was chosen and compared with a second refinement threshold value (*t*_2_). If the NCC score was greater than *t*_2_, the corresponding coordinates, scale and angle were taken as possible candidates for the refinement position matching process.

**Step 2.2 Robust angle measurement using rotation angle refinement matching.** To improve the accuracy of the PCB alignment process, the present study performed angle matching with a resolution of 0.1° in the refinement rotation matching process. In other words, for each angle candidate obtained in Step 2.1, a further matching process was performed around this angle candidate with a tolerance of $\pm \left(\frac{\alpha }{2}\right){}^{\circ}$ using an angular resolution of 0.1°, as shown in Figure 4d. In particular, search windows were created with center positions and sizes based on the position and scale candidates obtained in Step 2.1 and the refined angles described above. 

The NCC scores between the template (*T*) and each search window were computed and the orientation of the window which returned the largest NCC score was selected as the refined rotation matching solution. It should be assumed that for simplicity the angular resolution used in Step 1 was set as α = 10°. Thus, in Step 2.2, it was necessary to compare the similarity scores of the search windows with 100 different angles. A bilinear interpolation approach was employed to get the pixel values in the different rotation angles. The final outputs of the matching process were targets with the following parameters: angle (*θ*), scale (*s*) and coordinates (*x*, *y*).

## 5. Acceleration of the RST Template Matching Refinement Algorithm

To improve the matching time, in Step 1 of the RST refinement algorithm, the weak features of the search windows scanned across the test image were quickly rejected and a reduction of the operations in the matching process was then performed. Moreover, the entire matching process was performed on the GPU embedded in the NVIDIA TX2 Jetson development kit; thereby accelerating the processing time compared to that achieved using a CPU implementation.

### 5.1. Acceleration of Rotation Matching Using Quickly Rejecting Weak Features and Converting NCC Formula to Sum of Products 

#### 5.1.1. Quickly Rejecting Weak Features

As described above, in Step 1 of the rotation matching process, the average grayscale value of all the pixels along each radial line was calculated as the radial feature. It was found that, on the test images, there existed regions where the intensity values of the pixels were significantly different from the intensity values of the pixels on the template. As a result, the radial features on those regions were too different with the radial features on the template. We named those features the weak features. To accelerate the matching algorithm, when we scanned the radial features of the template on the test image, we converted the radial features on search windows and on the template into binary features. Then, we measured the distance between the two binary features. If that distance was greater than a certain threshold (*t_b_*), the corresponding search window was simply ignored. Otherwise, the NCC similarity scores between the radial features of the template and those of the search window were computed in the normal way, as described in Section 4. In this manner, we could reduce the number of similarity measurements in the scanning process. Through it, the scanning time was reduced. Converting from radial features to binary features was executed as follows:(7)$$b_{T}\left(i\right)=\{\begin{array}{c} 1,\;\;Rq_{i}\geq \mu_{Rq}\\ 0,\;\;Rq_{i}<\mu_{Rq} \end{array},\;$$
(8)$$b_{SW}\left(i\right)=\{\begin{array}{c} 1,\;\;Ra_{i}\geq \mu_{Ra}\\ 0,\;\;Ra_{i}<\mu_{Ra} \end{array}$$
*i* = 1, 2,…*Nr*,
(9)$$D_{b}=\left|\overset{N_{r}}{\underset{i=1}{\sum }}b_{T}\left(i\right)-\overset{N_{r}}{\underset{i=1}{\sum }}b_{SW}\left(i\right)\right|$$
where $b_{T}\left(i\right)\;and\;b_{SW}\left(i\right)$ are the binary features of a low-resolution template image and search window, respectively; and $D_{b}$ is the difference between the sum of $b_{T}$ and $b_{SW}$.

#### 5.1.2. Converting NCC formula to Sum of Products

After applying the quickly rejecting weak features method, the acceleration speed of the rotation matching method was further improved using our method described in [38]. The NCC formula in Equation (6) could be expanded as follows [39]:(10)$$\eta_{NCC}=|\frac{\sum_{i=1}^{Nr}(Rq_{i}*Ra_{i})\;-\;\mu_{Rq}\sum_{i=1}^{Nr}Ra_{i}\;-\;\mu_{Ra}\sum_{i=1}^{Nr}Rq_{i}\;+\;\sum_{i=1}^{Nr}\left(\mu_{Rq}*\mu_{Ra}\right)}{\sqrt{\sum_{i=1}^{Nr}{\left(Rq_{i}-\mu_{Rq}\right)}^{2}}\sqrt{\sum_{i=1}^{Nr}Ra_{i}{}^{2}-2\mu_{Ra}\sum_{i=1}^{Nr}Ra_{i}+\sum_{i=1}^{Nr}\mu_{Ra}{}^{2}}}|$$
where:(11)$$\sum_{i=1}^{Nr}Ra_{i}=Nr*\mu_{Ra},\;$$
(12)$$\sum_{i=1}^{Nr}Rq_{i}=Nr*\mu_{Rq},$$Therefore, Equation (10) can be rewritten as: (13)$$\eta_{NCC}=\left|\frac{\sum_{i=1}^{Nr}\left(Rq_{i}*Ra_{i}\right)-Nr*\mu_{Rq}*\mu_{Ra}}{\sqrt{\sum_{i=1}^{Nr}{\left(Rq_{i}-\mu_{Rq}\right)}^{2}}\sqrt{\sum_{i=1}^{Nr}Ra_{i}{}^{2}-Nr*\mu_{Ra}*\mu_{Ra}}}\right|$$

As shown in Figure 4a, in searching for the angle candidates at any location on the test image, it was necessary to compare the similarity between the radial lines on the template and the rotations of the radial lines in the search window. In other words, the *Nr* values in $Ra_{i}$ were rotated and their NCC similarities were computed with the corresponding values in $Rq_{i}$. Obviously, when rotating $Ra_{i}$, in Equation (13), $\mu_{Ra}$ is not changed. In other words, only the order of the elements in $Ra_{i}$ was changed. Moreover, in Equation (13), $\mu_{Rq}$ and $\sqrt{\sum_{i=1}^{Nr}{\left(Rq_{i}-\mu_{Rq}\right)}^{2}}$ can both be pre-computed and *Nr* is a constant. Therefore, in choosing the maximum similarity score between the radial features on the template and those on the search window, it was only necessary to consider the sum of the products ($\eta_{SoP}$) of $Rq_{i}$ and $Ra_{i}$, rather than the entire NCC formula. That is:(14)$$\eta_{SoP}=\sum_{i=1}^{Nr}\left(Rq_{i}*Ra_{i}\right)$$In this way, the operators in Equation (6) are significantly reduced to a sum of *Nr* products as in Equation (14).

After finding the search window that had a maximum similarity score, it was checked whether or not it was an angle candidate by calculating the maximum NCC score. Subsequently, that NCC score was compared with the threshold value, *t_1_.* The maximum NCC score can then be computed simply as
(15)$$\eta_{NCC}=\left|\frac{\eta_{SoP}-Nr*\mu_{Rq}*\mu_{Ra}}{\sqrt{\sum_{i=1}^{Nr}{\left(Rq_{i}-\mu_{Rq}\right)}^{2}}\sqrt{\sum_{i=1}^{Nr}Ra_{i}{}^{2}-Nr*\mu_{Ra}*\mu_{Ra}}}\right|$$

In this case, at each location of the scanning process in the rotation matching, the operations (addition, subtraction and division by standard deviations) of the similarity measurement are remarkably reduced. As a result, the matching time was improved.

### 5.2. Acceleration of RST Refinement Template Matching Algorithm by Running on Parallel Threads of GPU with Hash Tables

To further accelerate the matching process, the entire RST algorithm was embedded in the GPU of the NVIDIA Jetson TX2 kit. The GPU on the kit had two streaming multiprocessors (SM); each with 128 cores and four warp schedulers. In every instruction cycle, each warp scheduler selected one warp to execute. Each warp had a maximum size of 32 threads. In the present study, the RST algorithm was executed through a C++ CUDA program. The program was combined of C++ source code on the host and device. The host code referred to the code executed on the CPU and its memory. That code was used to declare and allocate memory on both the host and the device (GPU). The host code also transferred data from the host to the device. The device code (also known as the kernel) was implemented on the GPU and its memory, and it runs the RST algorithm in parallel through the threads on the GPU. The matching process was executed using the following steps. 

#### 5.2.1. Acceleration of Rotation Matching Using CUDA

First, a kernel was created to execute the angle matching process performed in Step 1 of the algorithm, in which the radial features of the low-resolution template (*T’*) were scanned over the test image. To reduce the matching time, the similarity measurement process at each location on the test image was distributed over the different threads of the GPU using a 2D grid. Referring to Figure 5 for illustration purposes, for a test image with a size of W = 640 (pixels) × H = 480 (pixels), each block is assigned 4 × 4 threads. Therefore, the total number of blocks in the 2D grid was equal to N = 160 (blocks) × M = 120 (blocks). 

#### 5.2.2. Location Refinement Matching Algorithm Using CUDA

The CUDA kernel for the location refinement matching step is shown in Algorithm 1. After converting the candidates to high-resolution images, each coordinate candidate was compared with its extension pixels ((*2^Np^*-1) pixels). In the kernel, the index of the threads was arranged along the *x* dimension of the 2D grid and was taken as the number of extension pixels; while the number of candidates was arranged along the *y* dimension. When implementing the refined position matching step in parallel on the threads of the GPU, the coordinates of the extension pixels were obtained using hash tables, with the thread index serving as the look-up keys. The hash table outputs indicated the *x* and *y* indexes of the extension pixels (see Figure 6). Then, we added those indexes and the coordinate candidate (after converting to high-resolution coordinate) to the refined coordinates.

The hash function of refined x coordinate is a modulo operator based on the index of the threads along the x dimension, *idxX*, as follows:*h_1_(idxX)* = *idxX***mod***2^Np^*(16)Similarly, the hash function of the refined y coordinate is a division as follows:*h_2_(idxX)* = *idxX***div***2^Np^*(17)With a coordinate candidate (x_Cand1_, y_Cand1_) from the angle measurement step of the RST refinement algorithm and its coordinate in a high-resolution image (*x_Cand1_* ∗ *2^Np^*, *y_Cand1_* ∗ *2^Np^*), the refined coordinates (x_Refined_, y_Refined_) of the extension pixels are calculated as
x_Refined_ = *h_1_(idxX)* + (*x_Cand1_* ∗ *2^Np^*)(18)
y_Refined_ = *h_2_(idxX)* + (*y_Cand1_* ∗ *2^Np^*)(19)The search window for Step 2.1 in the testing process of the RST refinement algorithm took the refined coordinates (x_Refined_, y_Refined_) as a center point.

#### 5.2.3. Rotation Angle Refinement Matching Algorithm Using CUDA

Algorithm 2 shows the CUDA kernel for the refined rotation angle matching step. The kernel was again implemented using a 2D grid with the index of the threads running along the *x* dimension taken as the index of the angles, and the index of the candidates was arranged along the *y* dimension. To obtain the required angular accuracy of 0.1°, the index of the threads along the *x* dimension was set in the range of (0, (α∗10)). The refined angle, *θ_Refined_*, is determined as follows:*θ_Refined_* = *idxX*/10.0 + *θ_Cand2_*(20)
where *θ_Cand2_* is the angle candidate obtained from Step 2.1 in Section 4. The refined angle was used as the orientation of the search window in Step 2.2 of the proposed RST testing.
**Algorithm 1** Pseudo-code of location refinement matching 1:**Inputs**: Test image *I*, size of template *w × h*, coordinate candidate ($x_{Cand1}$, $y_{Cand1}$), angle
candidates $\theta_{Cand1},$ scale candidates $s_{Cand1}$, pyramid level $N_{P}$, number of candidates $N_{Cand1}$2:**Outputs**: Correlation coefficient $\eta_{NCC}$3:**X index**: *idxX*←*blockDim.x* ∗ *blockIdx.x* + *threadIdx.x*   //*Number of extension pixels*4:**Y index**: *idxY*←*blockDim.y* ∗ *blockIdx.y* + *threadIdx.y*   //*Number of candidates*5:**if** (*idxX* < $2^{N_{P}}$) *and* (*idxY* < $N_{Cand1}$) **then**6: **Coordinate X**: $x_{Refined}\leftarrow \;$(*idxX*
**mod**
$2^{N_{P}}$) + ($x_{Cand1}$ + $2^{N_{P}}$)  //*Refinement coordinate x*7: **Coordinate Y**: $y_{Refined}\leftarrow$ (*idxY*
**div**
$2^{N_{P}}$) + ($y_{Cand1}$ + $2^{N_{P}}$)  //*Refinement coordinate y*8: **Scale s**: *s*
$\leftarrow s_{Cand1}$[*idxY*]9: **Angle**
θ: θ
$\leftarrow \theta_{Cand1}$[*idxY*]10: **for**
*j* in *h*
**do**11:  **for**
*i* in *w*
**do**12:   Collect intensity pixel values inside a search window with a center point at
   ($x_{Refined}$, $y_{Refined}$), an orientation: θ, and a scale: *s*13:  **end**14: **end**15: Calculate the NCC score $\eta_{NCC}$ between the template and the search windowend16:**end**17:**Return**$\eta_{NCC}$
**Algorithm 2** Pseudo-code of rotation angle refinement matching1:**Inputs**: Test image *I*, size of template *w × h*, coordinate candidate ($x_{Cand2}$, $y_{Cand2}$), angle
candidates $\theta_{Cand2},$ scale candidates $s_{Cand2}$, angular resolution α, number of candidates $N_{Cand2}$2:**Outputs**: Correlation coefficient $\eta_{NCC}$3:**X index**: *idxX*←*blockDim.x* ∗ *blockIdx.x* + *threadIdx.x*   //*Angular resolution*4:**Y index**: *idxY*←*blockDim.y* ∗ *blockIdx.y* + *threadIdx.y*   //*Number of candidates*5:**if** (*idxX* < α∗10) *and* (*idxY* < $N_{Cand2}$) **then**6: **Angle**
θ: $\theta_{Refined}$
$\leftarrow \left(idxX\;/\;10.0\right)+\theta_{Cand2}$[*idxY*]   //*Refined angle*7: **Coordinate X**: $x\leftarrow x_{Cand2}$[*idxY*]8: **Coordinate Y**: y←
$y_{Cand2}$[*idxY*]9: **Scale s**: *s*
$\leftarrow s_{Cand2}$[*idxY*]10: **for**
*j* in *h*
**do**11:  **for**
*i* in *w*
**do**12:   Collect intensity pixel values inside a search window using a bilinear interpolation approach with a center point at (*x*, *y*), an orientation: $\theta_{Refined}$, and a scale: *s*13:  **end**14: **end**15: Calculate the NCC score $\eta_{NCC}$ between the template and the search windowend16:**end**17:**Return**$\eta_{NCC}$

## 6. Experimental Results and Analysis

The experiments focused on two main aspects, namely (1) the matching time and accuracy of the proposed RST algorithm; and (2) the practical feasibility of the PCB alignment process. 

### 6.1. Data Collection

To support the RST algorithm testing process, two datasets were compiled, namely a fiducial marks dataset and a PCB component dataset. The former dataset was constructed using 30 fiducial mark templates collected from grayscale PCB images (as illustrated on the first row of Figure 7). Each fiducial template was tested on 20 fiducial PCB test images, where these test images comprised five blurred images, five Gaussian noise images, five rotated images and five scale images in the range of 0.8 to 1.2 with a scale interval of 0.1. The six hundred test images were equally divided into three different image sizes, namely 640 × 480 pixels, 800 × 600 pixels and 1280 × 960 pixels. 

The PCB component dataset was compiled using 20 templates and 400 test images. The templates consisted of various common PCB components, including integrated circuits (ICs), jacks and sockets (as illustrated on the second row of Figure 7); while the test images were captured directly from PCB samples using a CCD camera with a resolution of 1280 × 960 pixels. As for the fiducial marks dataset, the test images in the PCB component dataset contained an equal number of blurred, noisy, rotated and scale images. 

Moreover, we also used alphabet blocks to generate a dataset which consisted of 10 templates and 400 perspective and occluded test images with a resolution of 1280 × 720 pixels to test the robustness of the RST refinement algorithm. The alphabet blocks were rotated in a pitch and roll direction in a range of ±30°. A Secure Digital (SD) cards dataset was generated to test and compare the performance of the proposed RST algorithm with another method. To find suitable parameters for the PCB alignment process, we used a dataset of cross-hair marks. Table 1 summarizes the datasets used to evaluate the performance of the proposed RST algorithm.

### 6.2. Results and Analysis

In evaluating the performance of the RST algorithm, the parameters were set as follows. **Step 1:** angular resolution α = 10° (number of radial lines *Nr* = 36) and number of scales *Ns* = 5 (in range of 0.8 to 1.2 with a resolution of 0.1). **Step 2:** angular resolution α = 0.1°. The threshold value for rejecting the weak features in Step 1 was set as *t_b_* = 1/3*Nr*, while the threshold values for the candidate selection in Step 1 and Step 2 were set as 0.85 for the rotation matching process and 0.7 for the location refinement matching process and rotation angle refinement matching process.

The experiments were conducted on two different platforms, namely a PC with an Intel Core i7-6700 (Intel Corporation, Santa Clara, United States) and the embedded NVIDIA Jetson TX2 system described in Section 3.1. For the PC platform, the tests were performed to compare the performance of the original RST algorithm [23] (PC-RST) and the improved RST algorithm (PC-Improved RST), whereas the latter algorithm adopted the methods described in Section 5.1 for rejecting the weak features and cutting down operators of the NCC formula. In addition, the performance of the improved RST algorithm was also compared with the matching results of a fast screening RST template matching (FAsT-Match) approach in Liu et al. [32]. The performance evaluations on the embedded system considered three different RST algorithm versions, namely (1) the RST refinement algorithm running on a CPU (em-RST); (2) the RST refinement algorithm with an acceleration of the rotation matching (described in Section 5.1) running on a CPU (emCPU-Improved RST); and (3) the RST refinement algorithm with acceleration of the rotation matching running on a GPU (emGPU-Improved RST).

#### 6.2.1. Comparative Evaluation Experiments Using Fiducial Marks Dataset

This testing was implemented to compare the performances between the platforms and between the RST algorithm versions based on the fiducial marks dataset. Table 2 summarizes the performance results obtained in the various tests on the two different platforms. On the PC-based platform, in general, the PC-Improved RST algorithm was around 4.5× to 6.0× faster than the original algorithm [23], PC-RST. Furthermore, PC-Improved RST was faster than FAsT-Match when the algorithms were tested on images with sizes of 1280 × 960 pixels, while the accuracy was about 2.8% lower than that of FAsT-Match. The results showed that the proposed method reduced the matching time of the RST matching process, while having no significant effect on the accuracy of the matching results.

Regarding the embedded platforms, a comparison of the emCPU-Improved RST and em-RST showed that the former algorithm reduced the matching time by around 70.3%, 71.2% and 61.5% for test image sizes of 640 × 480, 800 × 600 and 1280 × 960 pixels, respectively. These results proved that the two methods mentioned in Section 5.1 were also effective when running on the embedded system platform. Compared to the FAsT-Match algorithm, the emGPU-Improved RST algorithm was faster when tested on test image sizes of 1280 × 960 pixels. On the other hand, as shown in Table 2, for all of the algorithms, the matching time generally increased with an increasing test image size. However, for the emGPU-Improved RST, the matching time varied only very slightly as the test image resolution increased since the matching process was performed in parallel on the GPU. Consequently, the scanning processes were executed concurrently on threads of the GPU, and as a result, the algorithm can cope effectively with the increase of the resolution of the test image. Comparing the matching times between emCPU-Improved RST and emGPU-Improved RST, it was found that the GPU-based platform reduced the matching time by 2.9× and 4.8× for test images with a size of 640 × 480 and 800 × 600 pixels, respectively. Moreover, the matching time was reduced by 12.2× for the image with a size of 1280 × 960 pixels. In other words, the performance advantage of the GPU-based platform increased with an increasing test image size. Compared to em-RST, performing the matching process on the GPU reduced the matching time by around 32× for a resolution of 1280 × 960 pixels. The GPU-based matching accuracy had a value of around 96%, irrespective of the test image resolution. Overall, the results confirmed that the acceleration of the RST algorithm achieved by running the algorithm in parallel on the GPU was not obtained at the expense of a reduction in the matching accuracy.

#### 6.2.2. Comparative Evaluation Experiments Using PCB Component Dataset

The performance of the emGPU-Improved RST template matching was further tested on the PCB component dataset. As shown in Table 3, the matching time was found to be 0.382s. In comparison with the testing result on the fiducial marks database, the accuracy was degraded slightly to 94.5%. Figure 8 shows the matching error ratios of the different types of test images. Specifically, the rotated images accounted for 34% of the total matching errors in the test process. As described in Section 4, angle matching with an angular resolution of 0.1° was performed on all the test images. The percentage of noise test images was 8% of the totally error, while the algorithm could deal well with the blurred images with 4% error. As shown in Figure 8, the scaled images accounted for 54% of the total errors (5.5%) obtained in the testing process. Figure 9 shows more details about the scale error contribution on the scale factors. The error mostly took place within scale factors 0.8 and 1.2. It was found that the algorithm performance was extremely sensitive to the scale of the test images, particularly with large-scale images. Figure 10 presents some illustrative scale-matching results obtained by the emGPU-Improved RST algorithm for test images with scales ranging from 0.8 to 1.2 in intervals of 0.1. Figure 11 presents some illustrative matching results for the blurred and noisy images. 

In order to estimate the rotation matching of the proposed method, we selected 10 templates from the PCB component dataset, and tested on 100 images. More specifically, every template was tested on 10 rotational images. Each test image was cropped to a size of 500 × 500 pixels with the template located at the center, and then rotated from 0.00° to 2.00° with a resolution of 0.2°. Table 4 shows the rotation performance of the proposed RST algorithm. In general, the angular errors fluctuated around 0.2°. The maximum angular error was 0.3°, while the average angular error was 0.107°. Moreover, we used a standard deviation of error (StD) to measure the errors of the angular results. The StD was calculated by using Equation (21). In the article [2], the authors also used a high-resolution angle matching for template matching to position the marking point on the PCBs. They obtained the StD of angle matching of 0.2°. In this study, as shown in Table 4, the maximum StD of the angular error was lower with a value of 0.074°. The results showed that the refinement process of the rotation angle matching contributed towards an effective and stable method to demonstrate the robust accuracy of the matching process.

The standard deviation of error is calculated as follows:(21)$$\mathrm{StD}=\sqrt{\frac{\sum_{i=1}^{N}{\left(x_{i}-\bar{x}\right)}^{2}}{N-1}}$$
where StD: standard deviation of error, *N*: the number of test samples, *x_i_*: test error of the i^th^ test sample, $\bar{x}$: mean of test errors on test samples.

#### 6.2.3. Comparative Evaluation Experiments Using SD Cards Dataset

The performance of the RST refinement algorithm was compared with a parallelized template matching approach on an embedded platform [25] (it should be noted that the approach presented in [25] used the SAD method, which has a significantly cheaper computation cost but is not as robust as the NCC, to measure the similarity). Based on the information of the dataset used in that article, we generated a similar dataset of SD cards. Our dataset consisted of five templates with a resolution of 100 × 80 pixels and 50 test images with a size of 640 × 480 pixels. As the approach in [25], we only considered the rotation and translation functions. The threshold values *t*_1_ and *t*_2_ were adjusted to 0.9 and 0.8, respectively. Table 5 shows the matching time of the two approaches. The results showed that the RST refinement algorithm was a little bit faster than the algorithm in [25]. Figure 12 illustrates the matching results with the different angle of targets.

#### 6.2.4. Performance Evaluation Experiments Using Alphabet Blocks Dataset

We tested the emGPU-Improved RST algorithm using the alphabet blocks dataset with perspective and occluded images. The test process obtained an average execution time of 0.126s and an accuracy of 93.3%. Figure 13 illustrates the matching results on the dataset. Those results show that the proposed algorithm can deal well with perspective and the occluded conditions of images. In addition, the algorithm was also tested on easily confused images. Templates were correctly matched when they were placed next to similar objects (as shown in Figure 13g,h).

#### 6.2.5. Cross-Hair Mark Detection Experiments Using *emGPU-Improved RST*


A further series of experiments was performed to investigate the trade-off between the matching time and the accuracy of the improved GPU-based RST algorithm. In performing the experiments, the proposed RST algorithm was tested on 100 fiducial cross-hair mark images. Moreover, the threshold value in Step 1 (*t*_1_) was varied in the range of 0.6 to 0.95, while that in Step 2 (*t*_2_) was adjusted in the range of 0.6 to 0.75. In addition, to evaluate the performance of the proposed PCB alignment method in practical manufacturing environments, the test images were added to with noise, foreign objects, stains or smudges to the target region, or by changing the light conditions from bright to dark. As shown in Figure 14, as the threshold values were increased, the accuracy and matching time reduced. Based on an inspection of Figure 14, the optimal values of *t*_1_ and *t*_2_ were determined to be 0.85 and 0.7, respectively (giving an accuracy of 97% and an average matching time of 0.437s). Figure 15a shows the template (with a size of 303 × 303 pixels) used to detect the cross-hairs. The cross-hairs were captured by the two cameras in the hardware system with a resolution of 1280 × 960 pixels. The matching results (see Figure 15b–i) showed that although the region of targets on the test images were significantly changed, the proposed RST algorithm could overcome this effectively. It proved that the proposed RST algorithm obtained practical effectiveness which could struggle with changes on the PCB surface. 

#### 6.2.6. PCB Alignment Experiments Using emGPU-Improved RST 

The PCB alignment experiments were performed using the threshold values described above and randomly-chosen positions of the PCB in the settle area of the system. Figure 16 shows the initial locations of the two PCB cross-hair marks, as observed by the two cameras in the experimental setup (it should be noted that the small red cross-marks show the set points in the FOVs of the two cameras). Table 6 summarizes the experimental results obtained for the 10 trials. To evaluate the alignment performance, the Euclidean distance formula was used to measure the distance between the coordinates of the cross-hair marks after the alignment process and the coordinates of the set points. The computed distance values (expressed in pixels) were then divided by the calibration value (ΔX = ΔY = 300 pixels/mm) to obtain the corresponding metric values. As shown at the foot of Table 6, the average distance errors for cross-hair mark 1 and cross-hair mark 2 were 36.4 μm and 27.6 μm, respectively. In comparison to the packaging standard [38], the largest distance error (36.4 μm) was equivalent to 9% of the smallest pin spacing (0.4 mm) of quad flat packaged (QFP) integrated circuits (ICs), 2.8% of the smallest pin spacing (1.27 mm) of the small-outline IC (SOIC) package ICs, and 1.4% of the pin spacing (2.54 mm) of dual in-line packaged (DIP) ICs. The comparison showed that the alignment error was still within an acceptable range to serve in the PCB manufacturing processes. 

## 7. Conclusions

This paper presented an improved RST template matching technique for PCB alignment in the assembly line manufacturing of electronic devices. It was shown that the template matching process achieved an accuracy of around 96% following the application of a refinement process to the original position and angle matching results. Furthermore, through a parallel implementation on a GPU embedded system, the matching algorithm achieved a matching time of just around 0.3*s* when applied to a test image with a size of 1280 × 960 pixels. The experimental results obtained in a series of PCB cross-hair mark matching trials showed that the algorithm was robust towards the effects of the illumination conditions and the presence of noise, foreign objects, stains and smudges in the target area. Moreover, the alignment process had a maximum positioning error of just 36.4μm. It did not have too great an effect on the accuracy of the PCB manufacturing processes. Future studies will aim to further reduce the positioning error by replacing the three motors in the hardware system with more precise motors in order to reduce the error when converting from the system coordinates to the number of motor pulses. In addition, although the algorithm provides an effective method for the alignment process, in some circumstances of large scales, it gives wrong results. A large-scale matching process will be taken into account to improve the accuracy of the matching process.

## Figures and Tables

**Figure 1 sensors-20-02736-f001:**
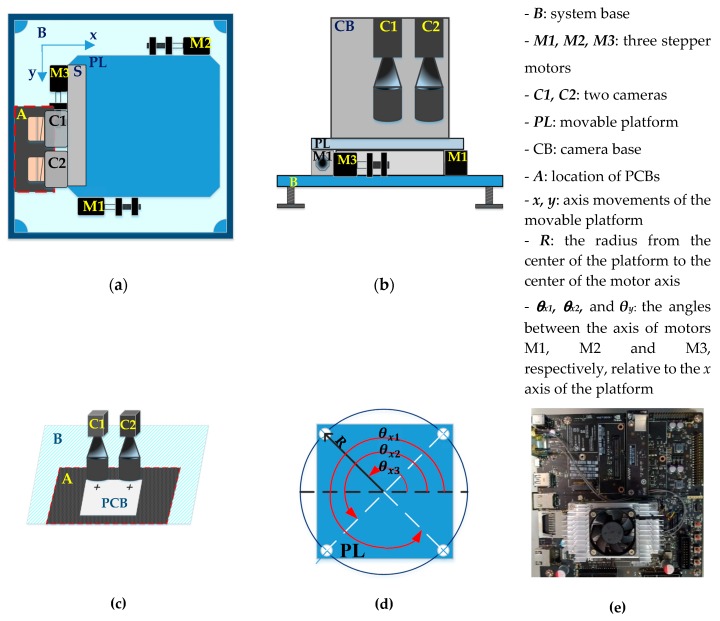
Overview of the hardware of the proposed embedded printed circuit board (PCB) alignment system. (**a**,**b**) are the top view and the front view of the vision alignment platform, respectively; (**c**) is the location of the PCB in the system; (**d**) is the description of the parameters of the movable platform (PL); and (**e**) shows the embedded-based controller using a NVIDIA Jetson Tx2 development kit.

**Figure 2 sensors-20-02736-f002:**
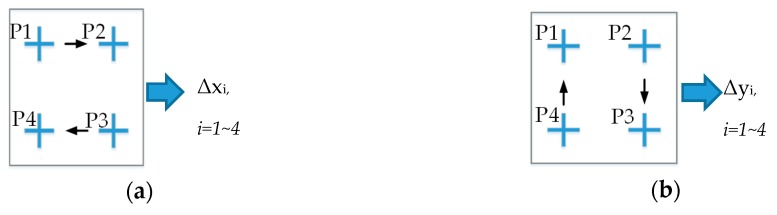
Motion of the cross-hair marks in the calibration process, in which, from P1 to P4, are reference points; Δxi, Δyi: are pixel distances between the two adjacent points along the *x* axis and the *y* axis, respectively. (**a**,**b**) show the travel of the cross-hair mark along *x* and *y* axis directions, respectively.

**Figure 3 sensors-20-02736-f003:**
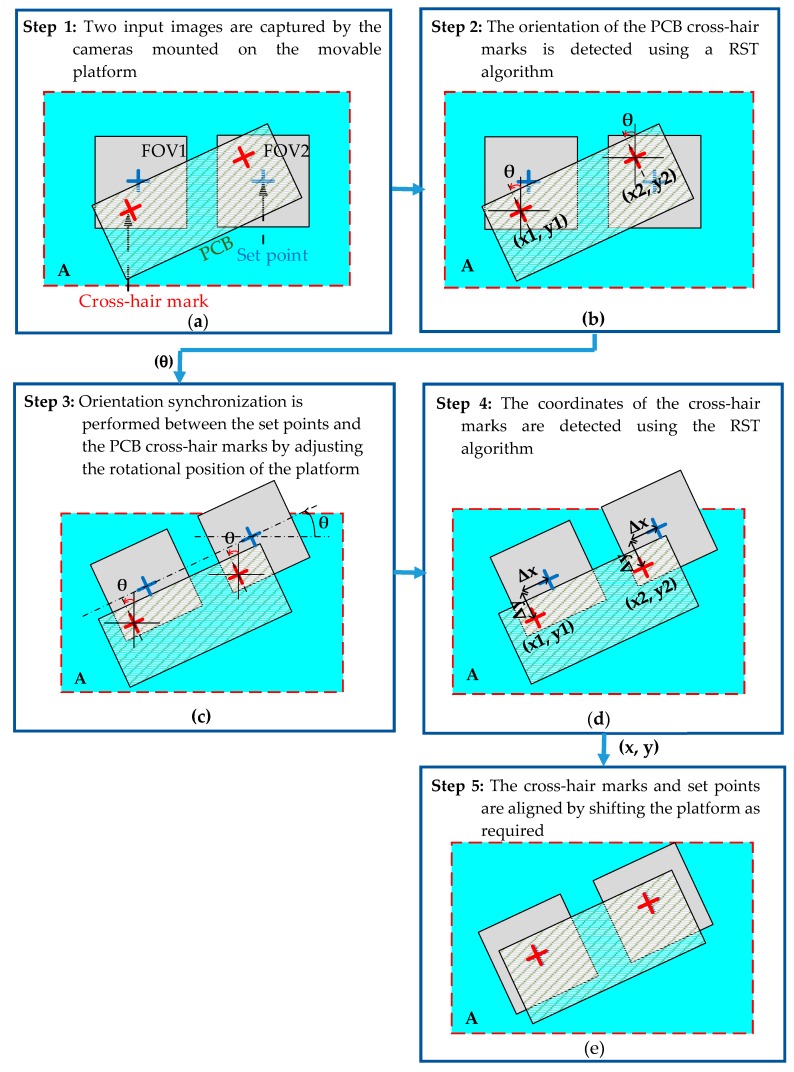
Global framework of the procedure of alignment between the marks on the PCB surface and the set points in the field of views (FOVs) of cameras; (**a**) initial positions of marks are detected after assigning new PCB. *FOV1* and *FOV2*: fields of view of cameras 1 and 2, respectively; A: settle area of PCBs. (**b**) Orientation angle of PCB is detected using the rotation, scale and translation (RST) algorithm; *θ*: rotation angle of mark. (**c**) Orientation synchronization is performed between the movable platform and the PCB. (**d**) Coordinates of marks are detected and the distances between the marks and the set points are computed; (*x1, y1*) and (*x2*, *y2*): coordinates of marks; Δ*x* and Δ*y*: distances between marks and set points; (**e**) Platform is translated in *x* and *y* directions to achieve alignment between the marks and the set points.

**Figure 4 sensors-20-02736-f004:**
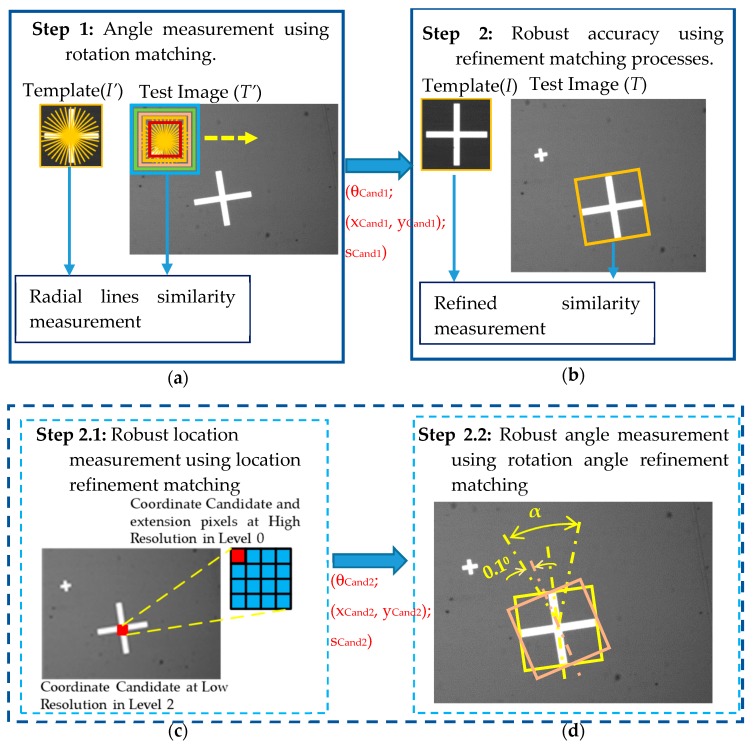
Global framework of the RST refinement template matching algorithm. (**a**) Angle matching process based on the radial line features, with outputs of the angle candidate (*θ_Cand1_*), the scale candidate (*s_Cand1_*) and the coordinate candidate (*x_Cand1_, y_Cand1_*); (**b**) refinement matching process comprising two sub-steps for the position matching and the angle matching, respectively; (**c**) refinement process of location matching with extension pixels after converting from a low-resolution image to a high-resolution image (figure shows illustrative case of pyramid up-sampling with level 2). The refinement process provides outputs of three parameters, namely angle (*θ_Cand2_*), scale (*s_Cand2_*) and coordinates (*x_Cand2_, y_Cand2_*); (**d**) refinement process of rotation matching fine-tunes the angle matching around the angle candidate (*θ_Cand2_*) with a tolerance of ±α/2 using angular resolution of 0.1°.

**Figure 5 sensors-20-02736-f005:**
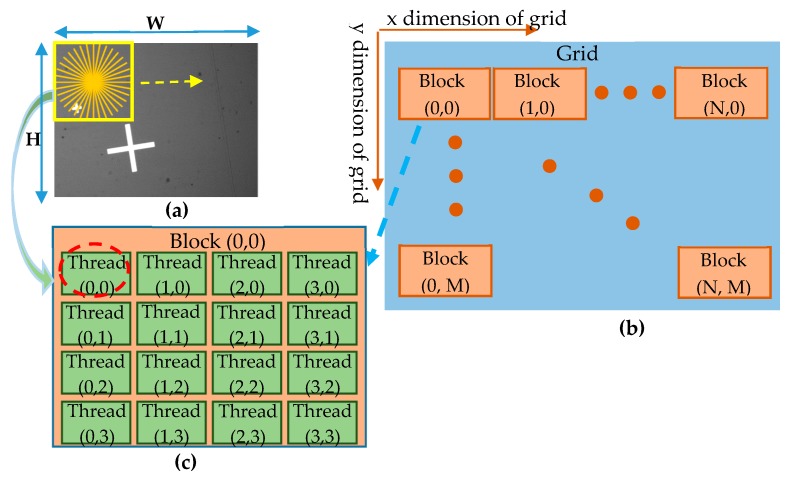
Structure of a 2D grid used to carry out the steps of the RST algorithm. (**a**) Scanning line of the radial features on the test image with a size of W × H; (**b**) structure of the grid with a size of N × M blocks; and (**c**) structure of the block with a size of 4 × 4 threads.

**Figure 6 sensors-20-02736-f006:**
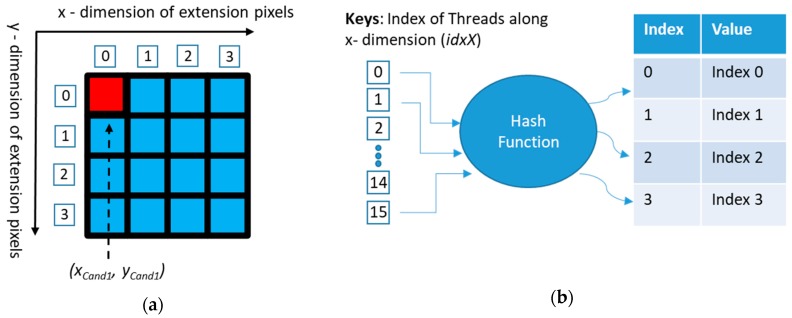
Finding index of the extension pixels using the hash table in the location refinement matching process running on the graphics processing unit (GPU) (figure displays a demonstration of the pyramid up-sampling with level 2, *N_P_* = 2). (**a**) Description of coordinate candidate (*x_Cand1_, y_Cand1_*) and extension pixels; (**b**) description of finding *x* and *y* indexes of the extension pixels.

**Figure 7 sensors-20-02736-f007:**
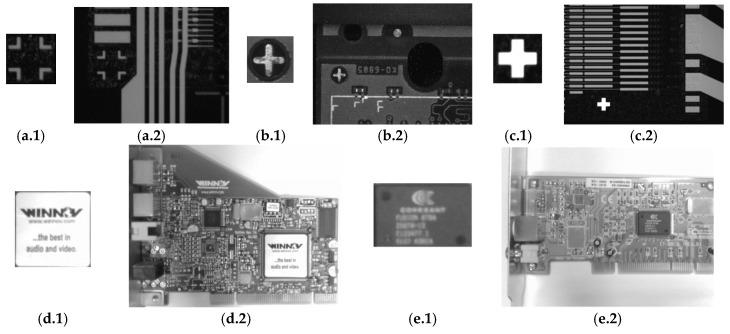
Illustrative examples of the templates and the test images used to evaluate the performance of the RST algorithms. Images in the first row are the template images and the test images in the fiducial marks dataset: (**a.1**,**b.1**,**c.1**) show template images with sizes of 137 × 137 pixels, 75 × 75 pixels and 129 × 129 pixels, respectively; while (**a.2**,**b.2**,**c.2**) show the test images with sizes of 640 × 480 pixels, 800 × 600 pixels and 1280 × 960 pixels, respectively. On the other hand, the images in the second row are the template images and the test images in the PCB component dataset: (**d.1**,**e.1**) show the template images with sizes of 257 × 257 pixels and 212 × 145 pixels, respectively; while (**d.2**,**e.2**) show the test images with a size of 1280 × 960 pixels.

**Figure 8 sensors-20-02736-f008:**
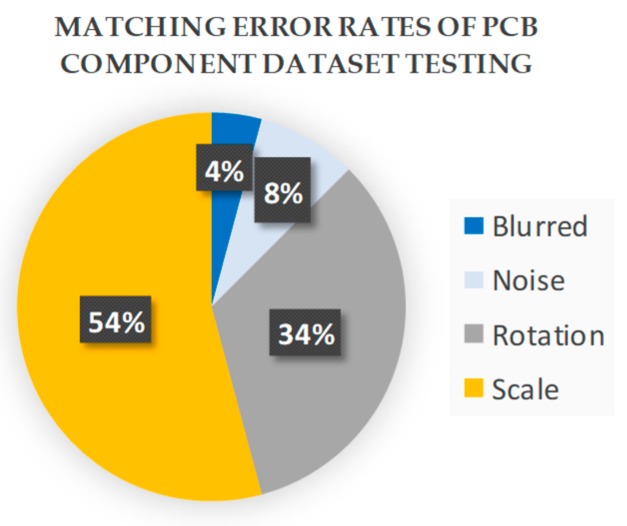
Matching error rates of the different test image types in the PCB component dataset.

**Figure 9 sensors-20-02736-f009:**
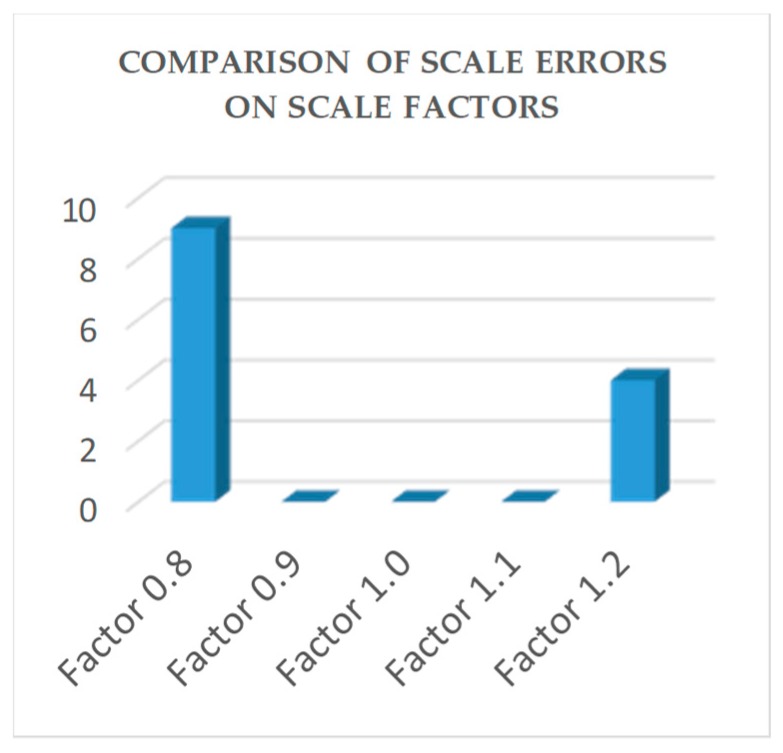
Comparison of the scale errors on 5 scale factors.

**Figure 10 sensors-20-02736-f010:**
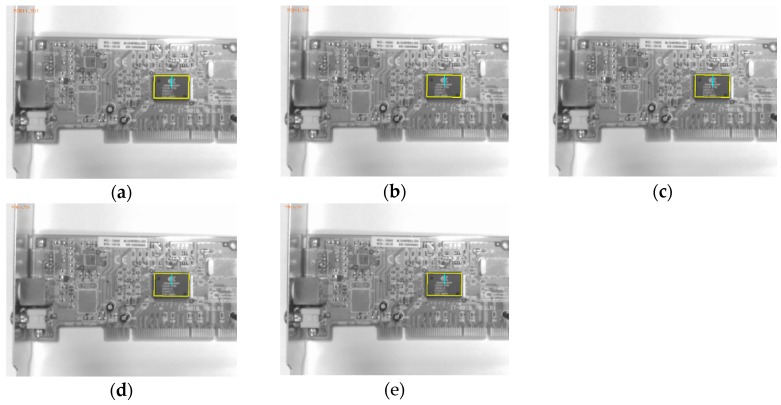
Results of the scale matching over the scale range of 0.8 to 1.2. (**a**–**e**) show the results for the test images (1280 × 960 pixels) with scales of 0.8, 0.9, 1.0, 1.1 and 1.2, respectively.

**Figure 11 sensors-20-02736-f011:**
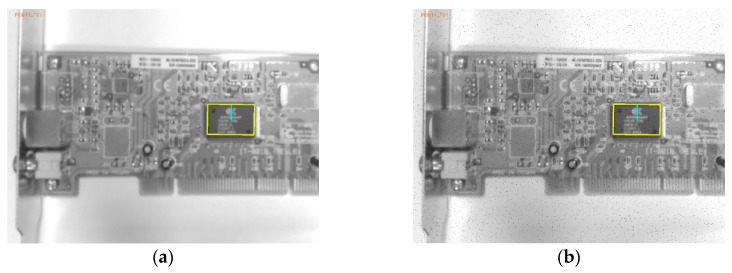
Results obtained for the test images with blurred and noise conditions: (**a**) blurred condition and (**b**) noise condition.

**Figure 12 sensors-20-02736-f012:**
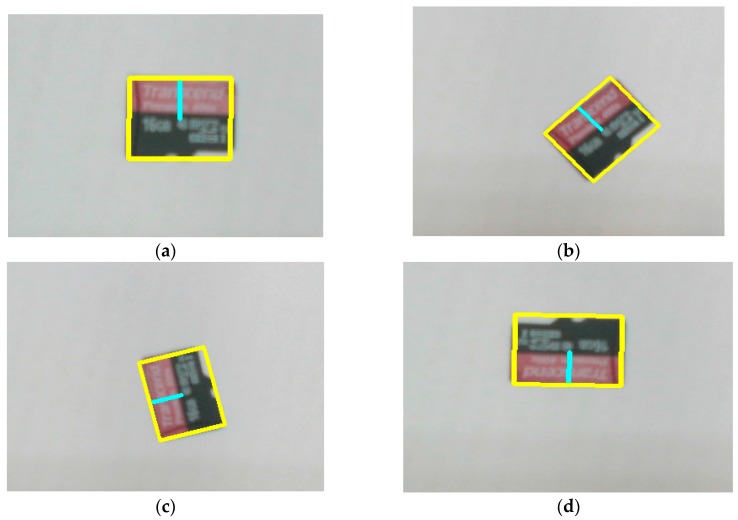
Matching results on the SD cards dataset: (**a**–**d**) show matching results with different angles.

**Figure 13 sensors-20-02736-f013:**
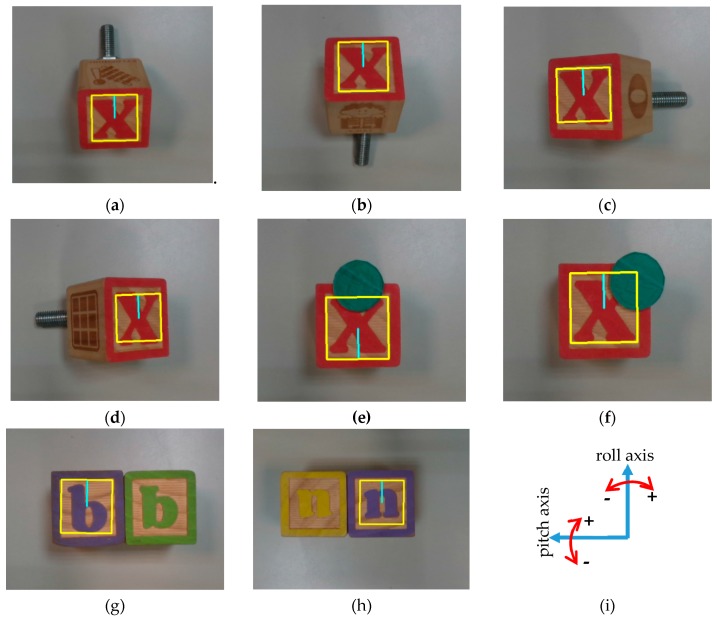
Matching results for the alphabet blocks dataset with the different perspective and occluded conditions: (**a**,**b**) show a description of the images with pitch angles of −30 and +30 degrees, respectively; (**c**) and (**d**) show a description of the images with roll angles of −30 and +30 degrees, respectively; (**e**,**f**) show a description of the occluded images; (**g**,**h**) show a description of the easily confused images, in which the former shows that the target “b” was matched correctly, while the object “inverted-q” was an easily confused object with target “b”. The same for the target “n” and the easily confused object “inverted-u” in (**h**); (**i**) illustrates the direction of pitch and roll axes. All the images have a resolution of 1280 × 720 pixels.

**Figure 14 sensors-20-02736-f014:**
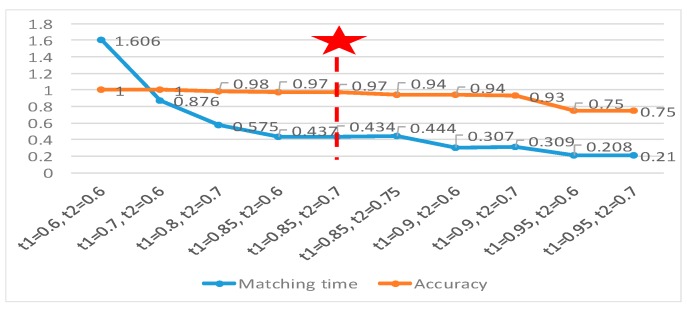
The matching time and the accuracy of a cross-hair mark testing on 100 test images.

**Figure 15 sensors-20-02736-f015:**
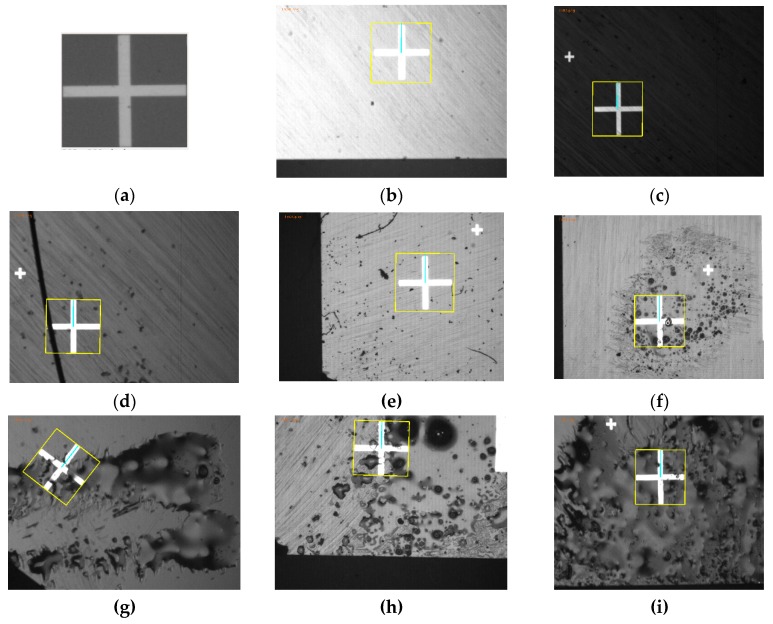
Matching results for the PCB cross-hair marks under different environmental conditions. (**a**) Cross-hair mark template (303 × 303 pixels). From (**b**) to (**c**), the test images have a size of 1280 × 960 pixels, in which: (**b**,**c**) are the matching results obtained under the bright and dark conditions, respectively; (**d**,**e**) are the matching results obtained for foreign object placement inside the targets; (**f**–**i**) are the matching results obtained for targets affected by stains and smudges.

**Figure 16 sensors-20-02736-f016:**
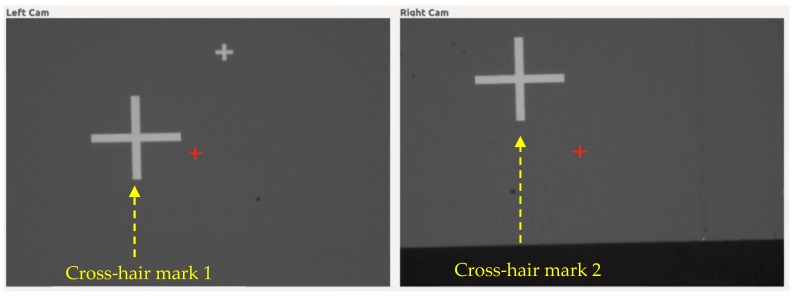
Fields of view of Cameras 1 and 2 showing the PCB cross marks (large white cross marks) and the set points on the fields of view (small red cross marks).

**Table 1 sensors-20-02736-t001:** Datasets are used to the evaluate performance of the RST algorithms.

Dataset	Size of Test Images (pixels)	No. of Templates	No. of Test Images
Fiducial Marks	640 × 480	10	200
	800 × 600	10	200
	1280 × 960	10	200
PCB Component	1280 × 960	20	400
Alphabet Blocks	1280 × 720	10	400
Secure Digital (SD) Cards	640 × 480	5	50
Cross-hair Marks	1280 × 960	1	100

**Table 2 sensors-20-02736-t002:** Comparison of the average matching time and the accuracy of the RST template matching algorithms on different platforms. On PC platform: the original RST [23] (PC-RST) and the improved RST algorithm (PC-Improved RST). On the embedded system: the RST refinement algorithm (em-RST), the RST refinement with acceleration of the rotation matching (emCPU-Improved RST) and the RST refinement with acceleration of the rotation matching running on GPU (emGPU-Improved RST). Those algorithms also were compared with a fast screening RST template matching [32] (FAsT-Match).

Methods	Image Size					
	640 × 480 (pixels)		800 × 600 (pixels)		1280 × 960 (pixels)	
	Time(s)	Accuracy	Time(s)	Accuracy	Time(s)	Accuracy
***PC-based Platform***						
PC-RST [23]	0.618	98.0%	1.075	98.0%	1.326	97.5%
FAsT-Match [32]	0.1	99.9%	-	-	0.4	99.8%
PC- Improved RST	0.099	97.5%	0.186	95.5%	0.291	97.0%
***Embedded System-based Platform***						
em-RST	1.914	97.0%	5.668	96.5%	9.517	95.0%
emCPU-Improved RST	0.568	92.0%	1.633	95.0%	3.664	98.0%
emGPU-Improved RST	0.197	96.5%	0.342	95.5%	0.301	96.0%

**Table 3 sensors-20-02736-t003:** The performance of the emGPU-Improved RST algorithm on the PCB component dataset.

Method	Average Matching Time(s)	Accuracy
emGPU-Improved RST	0.382	94.0%

**Table 4 sensors-20-02736-t004:** The rotational matching results is carried out on 10 rotation PCB images (500 × 500 pixels).

Test Images	Ground Truth (°)	0.0	0.2	0.4	0.6	0.8	1.0	1.2	1.4	1.6	1.8	Max. Error (°)	Mean Error (°)	StD
PCBR01	*Predict*	0.0	0.1	0.3	0.5	0.8	1.0	1.2	1.3	1.5	1.7			
	*Error*	0.0	0.1	0.1	0.1	0.0	0.0	0.0	0.1	0.1	0.1	0.10	0.06	0.052
PCBR02	*Predict*	0.0	0.2	0.4	0.6	0.8	1.0	1.2	1.4	1.6	1.7			
	*Error*	0.0	0.0	0.0	0.0	0.0	0.0	0.0	0.0	0.0	0.1	0.10	0.01	0.032
PCBR03	*Predict*	0.0	0.1	0.4	0.5	0.7	1.0	1.2	1.4	1.6	1.8			
	*Error*	0.0	0.1	0.0	0.1	0.1	0.0	0.0	0.0	0.0	0.0	0.10	0.03	0.048
PCBR04	*Predict*	0.2	0.3	0.5	0.7	0.9	1.1	1.3	1.5	1.7	1.9			
	*Error*	0.2	0.1	0.1	0.1	0.1	0.1	0.1	0.1	0.1	0.1	0.20	0.11	0.032
PCBR05	*Predict*	0.0	0.1	0.3	0.5	0.7	1.0	1.2	1.3	1.5	1.7			
	*Error*	0.0	0.1	0.1	0.1	0.1	0.0	0.0	0.1	0.1	0.1	0.10	0.07	0.048
PCBR06	*Predict*	0.0	0.1	0.3	0.5	0.8	1.0	1.2	1.3	1.5	1.8			
	*Error*	0.0	0.1	0.1	0.1	0.0	0.0	0.0	0.1	0.1	0.0	0.10	0.05	0.053
PCBR07	*Predict*	0.0	0.0	0.2	0.5	0.8	1.1	1.2	1.3	1.5	1.7			
	*Error*	0.0	0.2	0.2	0.1	0.0	0.1	0.0	0.1	0.1	0.1	0.20	0.09	**0.074**
PCBR08	*Predict*	359.7	359.9	0.1	0.3	0.5	0.7	0.9	1.1	1.3	1.5			
	*Error*	0.3	0.3	0.3	0.3	0.3	0.3	0.3	0.3	0.3	0.3	**0.30**	0.30	0.00
PCBR09	*Predict*	0.2	0.5	0.6	0.9	1.0	1.3	1.4	1.7	1.9	2.1			
	*Error*	0.2	0.3	0.2	0.3	0.2	0.3	0.2	0.3	0.3	0.3	**0.30**	0.26	0.052
PCBR10	*Predict*	359.9	0.1	0.3	0.6	0.7	0.9	1.1	1.3	1.5	1.7			
	*Error*	0.1	0.1	0.1	0.0	0.1	0.1	0.1	0.1	0.1	0.1	0.10	0.09	0.032
Average												0.16	0.107	0.04

**Table 5 sensors-20-02736-t005:** The performance of the emGPU-Improved RST algorithm on the SD cards dataset.

Methods	Average Matching Time on Each Type of Dataset(s)					Average Matching Time(s)
	SD1	SD2	SD3	SD4	SD5	
[25]	-	-	-	-	-	0.031
emGPU-Improved RST	0.025	0.024	0.025	0.043	0.023	0.028

**Table 6 sensors-20-02736-t006:** Experimental PCB alignment results. (Set point coordinates: in FOV1: (x1 = 598, y1 = 482) and in FOV2: (x2 = 480, y2 = 505).).

Experiment (Exp.)	Initial Coordinates (pixels)		Post-Alignment Coordinates (pixels)		Distance Error (μm)	
	Cross-Mark 1	Cross-Mark 2	Cross-Mark 1	Cross-Mark 2	Cross-Mark 1	Cross-Mark 2
Exp. #1	x1 = 427 y1 = 604	x2 = 308 y2 = 700	x1 = 608 y1 = 488	x2 = 308 y2 = 700	38.8	26.8
Exp. #2	x1 = 814 y1 = 580	x2 = 694 y2 = 697	x1 = 605 y1 = 471	x2 = 694 y2 = 697	43.5	58.2
Exp. #3	x1 = 426 y1 = 288	x2 = 301 y2 = 416	x1 = 608 y1 = 471	x2 = 301 y2 = 416	49.5	28.6
Exp. #4	x1 = 783 y1 = 295	x2 = 661 y2 = 408	x1 = 608 y1 = 478	x2 = 661 y2 = 408	35.9	21.1
Exp. #5	x1 = 755 y1 = 541	x2 = 632 y2 = 420	x1 = 602 y1 = 488	x2 = 632 y2 = 420	24.0	3.3
Exp. #6	x1 = 408 y1 = 283	x2 = 286 y2 = 432	x1 = 605 y1 = 478	x2 = 286 y2 = 432	26.8	23.6
Exp. #7	x1 = 500 y1 = 341	x2 = 378 y2 = 572	x1 = 595 y1 = 476	x2 = 378 y2 = 572	22.4	30.1
Exp. #8	x1 = 781 y1 = 551	x2 = 659 y2 = 666	x1 = 606 y1 = 482	x2 = 659 y2 = 666	26.7	18.8
Exp. #9	x1 = 644 y1 = 682	x2 = 524 y2 = 754	x1 = 611 y1 = 488	x2 = 524 y2 = 754	47.7	31.6
Exp. #10	x1 = 574 y1 = 227	x2 = 449 y2 = 325	x1 = 612 y1 = 478	x2 = 449 y2 = 325	48.5	34.3
Average Distance Error (μm):					36.4	27.6

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
