# Peer review of "A PCB Alignment System Using RST Template Matching with CUDA on Embedded GPU Boardâ€"

_sensors, 2020, doi:10.3390/s20092736_

Round 1
Reviewer 1 Report
The paper deals with the alignment process for Printed Circuit Boards (PCB). The alignment process is basically a computer vision process focused on the PCB alignment based on recognition and alignment of predefine marks.
The proposed procedure is based on a simplified version of the Normalized Cross-Correlation (NCC) and the parallel implementation of RST (Rotation, Scale and Translation) through the threads implementation approach over graphics Processing Unit (GPU). This GPU is really a GPU of a specific embedded system (Jetson TX2 of Nvidia).
More than research, the paper is a technical report of a methodological improvement of a computer vision issue with a strong application to the industry. The improvement is based on a reduced version of a known algorithm and the use of a top GPU board for its implementation.
The authors argue the paper contributions on three aspects, first and the second one is based on the computational cost of the NCC calculation, however, the improvement on computational cost is straight forward if you reduce the number of calculations in the NCC process. The third contribution is strictly related to the embedded board used for the implementation. The improvement is no coming from the research, it is coming from the embedded board (Jetson TX2) that allows a parallel implementation.
The related work: The related work is not focused on the contribution of the paper (basically is not related). Because most of the work mentioned is not using the same algorithm or the same or similar board (at least a GPU based board) and even some cases its are no applicable to the same problem. Just works [34, 35, 36, 37, and 39] show an average relevance because they have a similar approach problem, algorithms or they are using a similar board. The rest of the works named are far away from the paper proposal.
The proposed methodology, the algorithm, and the implementation approach are very well explained in sections 3, 4, and 5. However, for fashion approach, algorithm 1 and 2 descriptions should be done using latex on the paper, it is clear those are a picture attached with some resolution issues.
Experiment and results have big issues in the research approach paper.
The authors should perform and fair comparison of the proposed methodology, algorithm, and implementation, keeping a standard base. it means:
For NCC improvement comparison, it should be compared with similar algorithms that were developed for the same purpose, even including the NCC version without improvement proposed in this paper, all of them comparing the performance using the same embedded board.
For the algorithm implementation approach (contribution), the proposed version of NCC and RST should be compared in different board implementation.
These Ideas (performance comparison) aim to detach the relation of improvement algorithm and implementation in the case to visualize the real impact of the three contributions.
The comparison performed is unfair because some algorithms run on different boards and the algorithms that use a similar board have a very different approach.
Another comparative study that may be of interest and would highlight the contribution of the paper could be a comparison of the same proposed system implementation over the family of the jetson boards. It could be a fair competition and it could include a trade-off between financial cost and computational cost. Comparison of exactly the same proposed system could be done using: Jetson Nano, Jetson TX2, Jetson Xavier NX, and Jetson AGX
Author Response
Based on comments of Reviewer, I revised my manuscript. I summarize with points as follow and Please see the attachment for more details.
- The contributions were revised more clearly the physical meaning.
- The related work was revised and re-organized to more clearly and suitable with contributions.
- More test images were added and tested.
- The proposed algorithm was compared with other RST template matching algorithms and with parallelized embedded template matching paper.
- The pseudo-code format of Algorithm 1, 2 was modified.
Thank you.

Reviewer 2 Report
This paper presents an improved RST template matching technique for PCB alignment in the assembly line manufacturing of electronic devices. The authors build a Fiducial Marks dataset of 30 templates and 600 test images and a PCB Component dataset of 20 templates and 400 test images to evaluate the performance of the proposed method. The experimental results have shown that the template matching accuracy is 96%, the time cost is 0.3 s, and average alignment distance error is 36.4 μm. The technical methods of this manuscript are very clear. Three important comments are provided as follows:
1. The number of experimental images is slightly not enough to evaluate performance of the proposed method. It is suggested to add more images into the datasets.
2. There is a lack of comparison between the results of the proposed method and other methods. Some comparison with extant methods is suggested to be added if possible, in order to further highlight the robustness of the proposed method.
3. Mistyping errors should be corrected in the manuscript, e.g. Section 5.2.1 is reused and Section 6.2 title position is wrong.
Author Response
Based on comments of Reviewer, I revised my manuscript. I summarize with points as follow and Please see the attachment for more details.
- More test images with difference conditions, such as: perspective, occluded and easily confused images, were added and tested.
- The proposed algorithm was compared with other RST template matching algorithms and with parallelized embedded template matching paper.
- Mistyping errors were corrected.
Thank you.

Round 2
Reviewer 1 Report
I would like to thank the authors for taking into account my comments and suggestions to improve the research work carried out and strengthen the impact of this paper.
I understand that some of the requirements that I previously made cannot be carried out right now, however, I see that most of the corrections and suggestions were made in full, improving the paper and strengthening its impact.